# Green Technology Transfer, Environmental Regulation, and Regional Green Development Chasm: Based on the Empirical Evidence from Yangtze River Delta

**DOI:** 10.3390/ijerph192113925

**Published:** 2022-10-26

**Authors:** Yongmin Shang, Guoqing Lyu, Zefeng Mi

**Affiliations:** 1Institute of Ecology and Sustainable Development, Shanghai Academy of Social Sciences, Shanghai 200020, China; 2School of Resources, Environment and Architectural Engineering, Chifeng University, Chifeng 024000, China; 3School of Economics, Zhejiang University of Technology, Hangzhou 310014, China

**Keywords:** green technology transfer, environmental regulation, green development chasm, Yangtze River Delta

## Abstract

In the face of achieving the overall goal of emission peak and carbon neutrality, strengthening green technology transfer and environmental regulation is the key to narrowing the green technology gap and green development chasm between regions. This paper integrates green technology transfer, environmental regulation, and the green development chasm into one model, and analyzes the mechanism by which green technology transfer and environmental regulation impact the green development chasm. An empirical test was conducted by employing green technology transfer patent and panel data of the Yangtze River Delta from 2005–2019. The results are as follows: (1) Although the green development chasm still exists in the Yangtze River Delta, green technology transfer and environmental regulation have a positive impact on narrowing the regional green development chasm. Especially, the superposition of green technology transfer and environmental regulation can effectively make up for the lack of government and market regulation, and significantly promote the narrowing of the green development chasm. (2) Regional heterogeneity exists and developed regions can achieve the goal of narrowing the green development chasm by relying on green technology transfer or environmental regulation, while less developed regions must rely on the synergy of two dimensions. Thus, the coordination of green technology transfer and environmental regulation must be strengthened. Based on the above research, the main contributions of this paper are to analyze the theoretical mechanism of green technology transfer, environmental regulation, and regional green development chasm, to provide a theoretical and empirical basis for realizing the overall goal of regional green development, and suggestions for optimizing China’s current policies.

## 1. Introduction

In response to the global climate crisis, major countries and regions around the world have pledged carbon emission reduction targets [1]. Addressing climate change is a shared global responsibility; however, the gap of carbon emission levels and reduction capabilities among countries and regions in the world varies significantly, and the route to green development in less developed regions still faces enormous difficulties [2]. Therefore, how to enhance support to less developed regions and reduce regional green development chasm has become an important issue to achieve the common goal of global sustainable development. Green technology innovation is the key to addressing climate change and green development [3], and China has also proposed to promote a green and low-carbon technological revolution to realize carbon peaking by 2030 and carbon neutrality by 2060 [4]. Nevertheless, green technology innovation is characterized by multiple investment, high risks, long cycle as well as dual externalities, resulting in a lack of enthusiasm and weak ability for green technology innovation in less developed regions [5,6]. Meanwhile, green technology innovation activities are usually highly concentrated in a few economically developed regions [7], strengthening green technology transfer has become an important way to bridge the technological gap between regions and improve the overall level of green development [8].

In the 30 years since the United Nations Framework Convention on Climate Change was signed, green technology transfer has become a widely discussed topic in academia and a task that institutions and governments worldwide are committed to promote [9]. Yet, the global green development chasm is still obvious. According to World Bank data, five countries—Japan, the United States, Germany, South Korea, and China—accounted for 73% of the global low-carbon technology patent output during 2010–2015, whereas the rest of the countries accounted for merely 27% [10]. A large number of scholars have pointed out that green technology transfer is strongly market-oriented and characterized by incomplete competition and information asymmetry [11,12]. Under the paradigm of spontaneous cooperation, green technology transfer is prone to market failure and is usually highly concentrated in economically developed regions, with less developed regions struggling to gain adequate access to external green technologies, thereby further exacerbates the green development chasm [7]. For this reason, the relationship between green technology transfer and the green development chasm calls for more detailed consideration. In order to compensate for market failure, academics have addressed that the positive role of environmental regulation in stimulating green technology innovation, increasing the flow of green technology across regions, and bridging the green development chasm [13,14]. Realistically, however, the role of environmental regulation on the green development chasm is complex [13]. On one hand, some regions have chosen to weaken environmental regulations in pursuit of short-term economic growth, leading to the entry of highly pollution and carbon-emissions industries, and being locked into a destructive development path [15]. On the other hand, as China vigorously promotes an ecological civilization, some regional governments have imposed strict environmental regulations, but due to the lack of access to external green technology and advanced elements, they are caught in a development dilemma, and the green development chasm has remains difficult to overcome [16]. It is therefore clear that neither green technology transfer nor environmental regulation can effectively narrow the green development chasm between regions. Tackling climate change is a systematic project, which relies heavily on the coordination of green technology, government regulation and the other related fields [17]. Few existing studies yet have examined green technology transfer and environmental regulation in a systematic analysis framework.

Based on this, this paper attempts to respond to the following crucial but not yet well-answered questions: Does green technology transfer contribute to narrowing the green development chasm and what is the inherent mechanism? What is the impact of environmental regulation on the green development chasm, and how has it been combined with green technology transfer on the green development chasm? Meanwhile, this paper considers the high regional integration development, active green technology transfer and strong inter-city environmental regulation demonstration in the Yangtze River Delta, this paper employs the data of the Yangtze River Delta from 2005 to 2019 for empirical testing.

The possible contributions of this paper are as follows: firstly, this paper emphasizes the necessity of narrowing the green development chasm in the era of emission peak and carbon neutrality. Additionally, different from the existing fragmented discussions, this paper incorporates green technology transfer, environmental regulation, and green development chasm into one analysis framework, which is a meaningful supplementation for current theoretical research. Secondly, this paper focuses on the heterogeneous impact of green technology transfer and environmental regulation on the green development chasm, the Yangtze River Delta in China is selected and divided it into two kinds of regions, namely, the central region with relatively high level of economic development and the non-central region with relatively low level. Thirdly, combining the OECD ENV-TECH classification and incoPat patent database, the number of green technology patents transferred in different cities at different time periods was counted, and together with the statistical yearbook, the effects of green technology transfer and environmental regulation on the green development chasm were empirically tested.

The structure of this paper is as follows: Section 2 reviews the relevant literature. Section 3 analyzes the theoretical mechanism of green technology transfer, environmental regulation, and green development chasm. Section 4 introduces the model, variable measurement, and data sources. Section 5 provides empirical results and related discussions. Section 6 gives conclusions and policy recommendations.

## 2. Literature Review

### 2.1. The Characteristics and Influencing Factors of Regional Green Development Chasm

Generally, chasm is similar to the concepts such as imbalance and disparity, which is widely used in discussions on digital chasm [18], innovation chasm [19], governance chasm [20], and ideological chasm [21]. In the context of global climate crisis, the green development chasm has become a hot topic in academia to discuss the inequalities in the global green economy level, green technology, and green system [22,23]. Despite the active emphasis on “inclusive green growth” by governments and international institutions, global green development inequalities are still rising [24]. A large number of scholars have called for bridging the regional green development chasm, which is also related to the smooth realization of the common goals of addressing global warming and green development [25,26]. Many scholars have found that the green development chasm exists not only at the global scale between developed and underdeveloped countries [27,28], but also within a country or even within a region [29,30,31]. Scholars have also drawn different conclusions from the analysis of the changing trend of the green development chasm in different regions. For example, under the dual challenges of economic and environmental protection, the less developed countries and regions are facing enormous difficulties in green development [32]. However, at a national or regional scale, the green development gap is generally narrowing [33]. The academics have done a lot of research on the influencing factors of the green development chasm, and scholars generally believe that green development chasm arises as a result of a combination of factors including the level of green technology, knowledge linkages, socio-economic environment, knowledge absorptive capacity, and government environmental regulations, etc. [13,32,33].

### 2.2. The Relationship between Green Technology Transfer and Regional Green Development Chasm

Regarding whether technology transfer has contributed to the narrowing of the regional green development chasm, there are two opposing views in academia. Scholars who support the view believe that technology absorption is an effective way to achieve catch-up with developed regions, and that knowledge flow promotes the convergence of inter-regional technology gaps and brings about inter-regional economic convergence [8,34,35]. Some scholars have found that when low-carbon technology is fully shared between countries, technology transfer can reduce global cumulative carbon emissions by about 40% [35]. In particular, those countries that actively promote the green technology transfer will also generally improve their green technology abundance and green economic growth capacity [36,37]. Through green technology transfer, enterprises can create more opportunities for local promotion and application of new technologies and help further to promote local R&D and cross-regional transfer of green technology [38]. Scholars who hold opposing views argued that along with the flow of factors, there may be a divergence of technologies between regions [39], and market-oriented technology transfer struggles to reduce regional economic disparities [40]. In general, there are still few academic discussions on the relationship between green technology transfer and the green development chasm, and the existing studies are unable to provide a good explanation of whether green technology transfer can narrow the green development chasm.

### 2.3. The Relationship between Environmental Regulation and Regional Green Development Chasm

In terms of the relationship between environmental regulation and green development chasm, there is still some debate in the academia. Concretely, some scholars believe that environmental regulation will widen the green development chasm, and there are disparities in the responses of regions with different economic bases [41,42]. Less developed regions may experience higher costs and declining enterprise productivity when faced with external pressure from environmental regulations [43]. Economically developed regions, in contrast, can be more likely to avoid the negative impact, and environmental regulation will even promote the agglomeration of innovation factors towards to developed regions [44], which in turn lead to a widening regional development chasm. Meanwhile, there is also a large number of scholars who believe that environmental regulation will not lead to a widening of the green development chasm [13,45]. On the one hand, environmental regulation will inhibit the entry or force the exit of inefficient enterprises and improve regional production efficiency, and then will also promote the transfer of production factors to high-productivity enterprises, optimizing the efficiency of resource allocation [45]. On the other hand, national unified environmental regulation will increase the entry cost of polluting enterprises in less developed areas, and discourage their entry, thus improve the green total factor productivity (TFP), which will be conducive to narrowing the regional green development chasm on a long-term perspective [13]. In addition, the impact of environmental regulation on the green development chasm is also reflected in the environmental and social dimensions. Related studies have found that environmental regulation reduces the transfer of pollution between regions [46] and promotes the convergence of environmental efficiency between regions [47]. Meanwhile, some scholars have also discussed matters from the micro-level of enterprises [48,49]. Lean green and green management will motivate enterprises to adopt more environmentally friendly practices, adopt clean technologies, reduce resource consumption, and waste discharge, as well as waste in different links such as production, transportation, inventory, etc. In turn, they improve the sustainable performance of enterprises and enhance green competitive advantage [50,51], thereby breaking the green barrier. At the same time, environmental regulation also promotes the spatial spillover of ecological welfare, thereby narrowing the gap in ecological welfare between regions [52].

In summary, the existing literature are generally fragmented, mainly focusing on the relationship between green technology transfer, environmental regulation and green development chasm separately, and unconsciously neglecting to discuss the joint effect under one integrated analytical framework. Especially in the era of emission peak and carbon neutrality, the green development of a region increasingly depends on a combination of multiple factors such as green technology transfer and environmental regulation. However, no matter which dimension, factor flows or institutions, it is difficult to tackle the issues of narrowing the regional green development chasm. Additionally, if we only discuss their relationship with the green development chasm from a certain dimension, it may lead to theoretical separation and fragmentation of practical work. At the same time, the characteristics of the regional green development chasm and the role of green technology transfer and environmental regulation are still controversial. Due to the availability of data, most of the existing studies on green technology transfer employ the data such as talent flow and investment, and rarely use panel data for overall analysis. The role of green technology transfer and environmental regulation on the regional green development chasm needs to be further verified. Whilst China has made impressive achievements in green technology transfer, environmental regulation and green development in recent years, there is still relatively little empirical research in academia on China and urban agglomerations particularly. Based on this, this paper constructs a theoretical model to investigate the mechanism of green technology transfer, environmental regulation, and the green development chasm, and employs green technology patent transfer data, panel data and spatial econometric models to test the effect of green technology transfer and environmental regulation on the green development chasm.

## 3. Theoretical Mechanisms

The relationship between technological progress and regional economic development is one of the most concerned issues in neoclassical economic theory [53]. In the era of the knowledge economy, inter-organizational and inter-regional knowledge spillover has become an overwhelmingly vital innovation factor for economic growth, whereby Huggins and Thompson (2014) proposed the concept of network capital to analyze the relationship between the innovation network formed by knowledge spillover and regional economic growth [54]. Green technology innovation is highly dependent on external knowledge spillovers [55]. Under the goal of green development, a large number of regions have introduced external technologies to improve their local green technology innovation capability, promote green economic growth and narrow the green development chasm with developed regions.

Green technology transfer is an important way to optimize the allocation of green technology resources and promote green development [8]. Since the cross-regional flow of green technology has overlaid effects of polarization and diffusion, it will also have a dual impact on the green development chasm. On one hand, green technology transfer can not only promote the flow of green technology elements, technology accumulation and technological progress in a relatively short period of time, but also stimulate green technology innovation where technology is transferred. Furthermore, through the process of “introduction-absorption-diffusion-re-innovation”, the less developed regions realize the optimization of technology structure and the improvement of green technology innovation capability [56,57]. In addition, green technology transfer will promote the formation of sound innovation environment, enabling both sides of the technology transfer to be embedded in the green technology innovation network, access to external green carbon technology and indirectly promote green technology progress [5,58]. As a result, less developed regions will narrow the green technology and development chasm with developed regions. On the other hand, green technology transfer is a kind of market behavior; driven by the spontaneous cooperation model and innovation agglomeration, green technology tends to cluster in regions with high benefit on investment, and developed regions will have a “siphon effect” on green technology [7]. On the contrary, the underdeveloped regions will be unable to fully obtain external green technologies and fall into low-tech lock-in and high-carbon development lock-in [59]. In turn, green technology transfer may exacerbate the green technology gap and development chasm between regions. Therefore, the key for green technology transfer to narrow the green development chasm is to effectively lead the transfer of green technology to less developed areas.

Environmental regulation is an important factor in green development, which has the dual impact on the green development chasm. On the one hand, economically developed regions generally have stronger environmental regulation than less developed ones [60], which may lead to the transfer of pollution activities across regions [61], and the agglomeration of factors such as green technology resources in developed regions with higher rates on investment. In turn, this may exacerbate the difficulties of green development in less developed regions. On the other hand, environmental regulation will also promote underdeveloped regions to catch up with green development. Strict environmental regulation will promote the transfer of production factors from low-productivity to high-productivity enterprises [62], and inhibit the entry of low-efficiency and polluting enterprises in underdeveloped areas, or force them to transform, and consequently significantly improve the level of industrial green development and TFP in underdeveloped regions, so as to narrow the green development chasm with economically developed regions [13]. At the same time, as all parts of China are paying increasing attention to environmental issues, environmental regulations have been strengthened as a whole, and regional differences have been continuously narrowed and even tended to be unified [63]. This makes less developed regions need to pay higher margin pollution costs than developed regions, creating a disincentive for polluting enterprises and pushing underdeveloped regions to speed up industrial green transformation [13], thereby narrowing the green development chasm. Nevertheless, the impact of environmental regulation on the green development chasm is not only affected by the intensity of environmental regulation, but also determined by the type of environmental regulation. According to the policy of government intervention, environmental regulation can usually be divided into command-and-control type and market-incentive type [57]. The former refers to the formulation of legislation and standards, while the latter involves grants, subsidies, or tax reduction and so on [64]. Although the academia has discussed the impact of different types of environmental regulations on green technology innovation and environmental efficiency [65,66], some scholars believe that different types of environmental regulation tools have unique advantages, and “carrots” and “sticks” are usually interactive and irreplaceable [67]. Additionally, the series of criteria and standards for cohesion is commonly employed as the architecture of environmental regulation.

Environmental regulation not only directly affects the green development chasm, but also affects the green development chasm by affecting green technology transfer. Strict environmental regulation will not only stimulate green technology innovation, but also increase the cross-regional flow of green technology [14], the rate of green technology diffusion in areas is usually higher with strict environmental regulations than the loose ones, and environmental regulations will create an inductive effect on the green technology transfer [68]. Green technology transfer and environmental regulation emphasize market forces and government behavior, respectively, will promote the efficient allocation of green technology resources, while environmental regulation and other supporting policies will make up for the failure of market regulation and facilitate the transfer of green technology to underdeveloped areas. Evidently, green technology transfer and environmental regulation interact with each other and jointly promote the narrowing green development chasm between regions [13]. It should be pointed out that green technology transfer and environmental regulation are both complex systems, and their effects on the regional green development chasm are influenced by a variety of factors, especially enterprise green management level and local regional environmental factor, such as economic level, industrial characteristics, government regulation, and technological absorptive capacity and so on [69]. These factors affect whether green technology can be transformed into local green economic growth capacity and whether green development can be achieved to catch up and thus narrow the green development chasm. The theoretical mechanism is shown in Figure 1 below.

## 4. Models and Methods

### 4.1. Model Settings

Based on neoclassical growth theory, this paper introduces network capital theory and constructs a spatial econometric model that can consider the spatial relevance of economic activities to examine the effect of green technology transfer and environmental regulation on the green development chasm.

Spatial econometric models commonly used in academia include the Spatial Error Model (SEM), Spatial Autoregressive Model (SAR), Spatial Durbin Model (SDM) and Spatial Cross Model (SAC). Different spatial econometric models assume different spatial conduction mechanisms, and their economic implications are also different; concretely, the SEM model assumes that green technology transfer and environmental regulation are caused by random shocks, and their spatial effects are mainly transmitted through the error term. Additionally, the SAR model assumes that the explained variables will affect the green development chasm in other regions through spatial interaction. The SAC model and the SDM model consider both types of spatial transmission mechanisms, and the SDM model also considers the spatial interaction, that is, the green development chasm in several regions is not only affected by local independent variables, but also by the green development chasm and independent variables in other regions. Therefore, the choice of spatial econometric model is crucial. Given the different meanings revealed by different models, to obtain the best fitting effect, this paper sets the model according to the path of OLS-SAR/SEM-SAC-SDM, in which Equations (1) and (2) are the SDM model and the SAC model, respectively, and Equations (3)–(5) are the SAR model, the SEM model, and the OLS model.
(1)$$\ln GDC_{it}=\beta_{0}+\delta W\ln GDC_{it}+\beta_{1}\ln GTT_{it}+\beta_{2}\ln ER_{it}+\beta_{3}\ln control_{it}\;+\theta_{1}\ln GTT_{it}+\theta_{2}\ln ER_{it}+\theta_{3}\ln control_{it}+\varepsilon_{it}$$
(2)$$\ln GDC_{it}=\beta_{0}+\delta W\ln GDC_{it}+\beta_{1}\ln GTT_{it}+\beta_{2}\ln ER_{it}+\beta_{3}\ln control_{it}+\mu_{it}\;\mu_{it}=\lambda W\mu_{it}+\varepsilon_{it}$$

When the spatial interaction investigated by the SDM model does not exist, and there is only one-way spatial correlation between regions, that is $\theta_{i}$ = 0 (i = 1, 2, 3), or when the coefficient of the spatial error term in the λ spatial SAC model = 0, it is The corresponding spatial SAR model:(3)$$\ln GDC_{it}=\beta_{0}+\delta W\ln GDC_{it}+\beta_{1}\ln GTT_{it}+\beta_{2}\ln ER_{it}+\beta_{3}\ln control_{it}+\varepsilon_{it}$$

When the spatial interaction coefficient $\theta_{i}$, the dependent variable spatial lag term coefficient δ and the regression coefficient in the SDM model satisfy $\theta_{i}$ = $-\delta \beta_{i}$, or when the coefficient of the spatial lag term δ in the SAC model satisfy δ = 0, it is the corresponding spatial SEM model:(4)$$\ln GDC_{it}=\beta_{0}+\beta_{1}\ln GTT_{it}+\beta_{2}\ln ER_{it}+\beta_{3}\ln control_{it}+\mu_{it}\;\mu_{it}=\lambda W\mu_{it}+\varepsilon_{it}$$

The OLS model does not consider the spatial correlation between regions, so when all coefficients of the spatial terms in the above models are 0, the corresponding OLS model can be obtained:(5)$$\ln GDC_{it}=\beta_{0}+\beta_{1}\ln GTT_{it}+\beta_{2}\ln ER_{it}+\beta_{3}\ln control_{it}+\varepsilon_{it}$$

$GDC_{it}$ is the explained variable, indicating the green economy development gap index. $GTT_{it}$ is green technology transfer, $ER_{it}$ is environmental regulation, and $control_{it}$ is a series of control variables. W is a 41 × 41 spatial weight matrix constructed by straight-line distance between cities, and the matrix is standardized. $\mu_{it}$ and $\varepsilon_{it}$ are disturbance terms that obey the same distribution. The formula of W is:(6)$$W={\tilde{w}}_{ij}/\sum_{j=1}^{n}{\tilde{w}}_{ij}$$

In the formula, n is the number of rows and columns of the matrix, the number of cities at the prefecture level is 41. ${\tilde{w}}_{ij}$ is the spatial distance between the city i and city j in the matrix W.

### 4.2. Variable Selection

#### 4.2.1. Explained Variables

The measurement of regional development gap mainly include the Gini coefficient, Theil index, weighting coefficient, etc. [40], but it is difficult to describe by panel data. There are also a large number of scholars who directly use the per capita GDP to measure the regional development gap [40,70]. Pang [71] and Zhang [72] measured the gap between each region and the highest level of economic development in the same period, which has good applicability. The Green Economy, published by the United Nations Environmental Programme in 2011, states that a green economy means better use of natural resources, more sustainable growth, and a more active role in promoting economic development and environmental protection than traditional style. This paper mainly emphasizes the continuous reduction in carbon emissions and its environmental impact while the economic growth maintains stable, and thus adopts the economic scale created by unit carbon emissions to measure the level of green economic development, this paper hence defines the Green Development Gap Index (GDG) with reference to relevant research, and the calculation method is shown in Formula (7). As shown in Figure 2, from 2005 to 2019, the green development gap in the Yangtze River Delta showed a “W”-shaped fluctuation. Additionally, since 2014, the average index has generally shown an upward trend. The comparison and analysis of the total GDP per unit carbon emission of the cities in the Yangtze River Delta through the Theil index also shows similar characteristics, which means that the green development gap between cities in Yangtze River Delta has widened in recent years.
(7)$$GDG=\max({GDP_{ik}/CO2}_{ik})/\left({GDP_{ik}/CO2}_{ik}\right)$$

In the formula, $GDP_{ik}$ indicates the total GDP of city i in year k. $CO2_{ik}$ Indicates the carbon dioxide emissions of city i in year k.

**Figure 2 ijerph-19-13925-f002:**
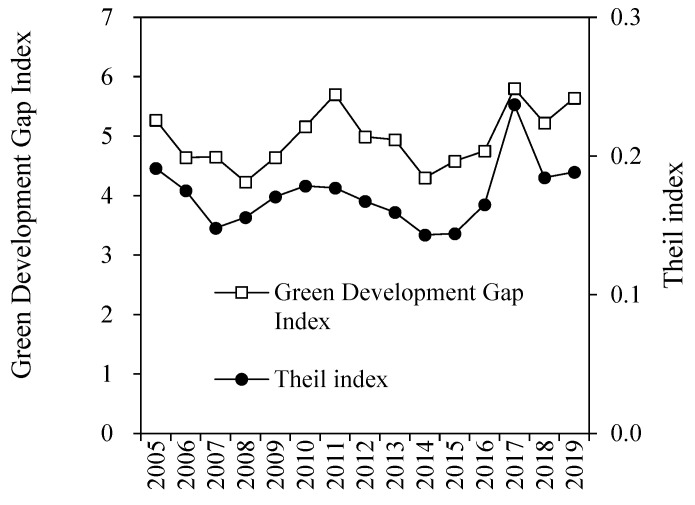
The average value of the Yangtze River Delta Green Development Gap Index in 2005–2019.

#### 4.2.2. Explanatory Variables

(1) Green Technology Transfer (*GTT*). There are various ways to measure technology transfer, such as technology exchange, talent flow, R&D cooperation, trade in goods and services, foreign investment, and technical assistance, of which patent transfer is essentially an interactive innovation process based on knowledge flow and innovation resource integration, both embedded into social network, which can reflect the sharing and transfer of technical knowledge among organizations and become an important data source for research on innovation networks [73]. Considering that transfer-in of patent is more reflective of local green technology innovation capabilities, this paper mainly uses the number to characterize green technology transfer. From Figure 3, the transfer of green technology in the Yangtze River Delta is highly agglomerated, and the amount of green technology transfer in a large number of cities was zero or slightly higher, which are mostly the economically underdeveloped cities in northern Jiangsu, Anhui, and southwestern Zhejiang, which are relatively marginalized. However, only a few cities had more than 100 green technology transfers, and the green technology transfer in the Yangtze River Delta was highly concentrated in economically developed cities such as Shanghai, Nanjing, Hangzhou, and Suzhou.

(2) Environmental Regulation (*ER*). There are four types of environmental regulation measurement methods: the first one is the single indicator approach, which measures the intensity of environmental regulation using individual indicator such as environmental governance input [74], the number of environmental violations [75], emission levels of individual pollutants [76], etc., is prone to research bias by a single indicator and usually lacks city-scale panel data. The second is the value-based scoring method, in which is environmental regulations are scored according to certain rules, but the method is relatively subjective. The third is classification inspection method, that is, based on different environmental regulation means and subjects to measure, such as environmental regulation is divided into command-and-control, market-incentive, public participation type, etc.; this method of urban data collection is usually more difficult to obtain. The last one is the comprehensive index method, which is calculated based on a number of indicators such as sulphur dioxide removal rate, industrial dust removal rate and industrial wastewater discharge compliance rate (Zhong, 2015; Li, 2018), alternatively, a system of evaluation indicators can be constructed based on a combination of indicators from multiple perspectives, which is more widely used. Hence, following the research of Zhong Maochu [77] and Li Hong [78], this paper adopts industrial sulfur dioxide, industrial smoke and dust, as well as the comprehensive utilization rate of general industrial solid wastes, decontamination rate of urban refuse, and the centralized treatment rate of sewage treatment plants. Due to the lack of data on urban-scale removal rate of industrial sulfur dioxide and soot, this paper uses industrial sulfur dioxide emissions per unit of industrial added value and industrial soot emissions per unit of industrial added value instead.

#### 4.2.3. Control Variables

Referring to relevant research, this paper mainly selects the following control variables: ① GDP per capita (*PGDP*). Numerous studies have shown that there is a significant positive correlation between the amount of green technology innovation cooperation and the level of economic development. According to the data of the Carbon Dioxide Information Analysis Center (CDIAC), as the world’s GDP per capita grows at an average rate of 2–3% per year, the global greenhouse gas also keeps increasing. This paper draws on Peng [61] and uses per capita GDP as a control variable. ② Technology absorptive capacity (TAC) is the ability to identify, absorb and utilize knowledge [79], this paper refers to the research of Pigato [10] to characterize the local green technology absorptive capacity using green technology patent stock. ③ Industrial structure (*IS*). Green technology transfer, environmental regulation and green economic growth are affected by the industrial structure of technology transfer place, taking reference to He [80], the paper deeply agree that the secondary industry is the main source of carbon emissions, the higher its proportion, the stronger the demand for green technology transfer, therefore the proportion of the secondary industry is chosen as the third control variable. ④ Government Science and Technology Investment (*GSTI*), strong government regulation can increase the demand for green technology transfer to promote innovation and technology transfer. However, government regulation can also increase public investment in green technology innovation, reduced private investment, and then hinder technology transfer [10]. In this paper, the proportion of government investment in science and technology to fiscal expenditure is used as a proxy for government investment in science and technology, which reflects the strength of the local government’s policy support for technological innovation and its role in directing enterprises to increase their R&D in green technologies. ⑤ Energy consumption (*EC*) is a key factor affecting green economic growth and carbon emissions. However, the statistical data of energy consumption in China’s cities over the years is incomplete, while the electricity consumption data is more complete, and the consumption data automatically recorded by electricity meters is more accurate [81]. Hence, electricity consumption can better represent the overall energy efficiency of China [82]. In the meantime, the GDP elasticity of electricity demand is very close to that of total energy. By comparing the logarithm of electricity consumption and energy consumption in various provinces and cities in China from 2005 to 2019, the correlation coefficient reaches 0.892. Therefore, this paper draws on the researches of Lin Boqiang et al. [81], Qin Bingtao et al. [82], Li Jianglong et al. [83], and uses electricity consumption data as a measure of energy consumption. The specific variable selection and corresponding explanation appears in Table 1.

To reduce the heteroscedasticity, each indicator was first logarithmically processed, and the following formula is used due to the existence of zero values for individual indicators:lnx=ln(1+x)

In the formula, lnx represents the value of the logarithm of x, x represents the data that needs to take logarithm. The correlation analysis was then carried out, and each indicator was correlated at the 1% significant level. The descriptive statistics of each variable are in Table 2.

### 4.3. Data Description

The patent data in this paper was obtained from the incoPat Global Patent Data Service website. Firstly, the CPC patent classification number of ENV-TECH released by OECD was used to obtain basic information such as green technology patent classification, applicant, address, patent name, application date, legal information event, etc. Then, patent in the Yangtze River Delta were extracted, and abnormal data such as information on individual patent applications and unknown information were excluded. Since there is an 18-month review period from application to grant, to ensure data integrity, this paper mainly extracts green technology transfer data from 2005 to 2019. Secondly, according to the patent event to extract the assignment information, if there are more than one participant in the patent, it is considered that there is an assignment connection between multiple assignments and assignees. For example, if the assignor is A and the assignees are B and C, it is considered that there is an assignment relationship between A and B and between A and C. Thirdly, based on the matching of the postal code of each city with the address of the applicant, the geospatial information is obtained. Moreover, the carbon emission data was obtained from the Carbon Emission Accounts & Datasets (CEADs), and the missing data are fitted according to the County-level CO_2_ Emissions in China of CEADs or the average growth rate in the past three years. Additionally, the other socio-economic data come from the China City Statistical Yearbook, city statistical yearbooks and statistical bulletins, etc.

## 5. Results

This section may be divided by subheadings. It should provide a concise and precise description of the experimental results, their interpretation, as well as the experimental conclusions that can be drawn.

### 5.1. Unit Root Test

In order to avoid pseudo-regression problems in the regression model, the LM test was chosen to test the unit root, the purpose of which is to test whether the indicator data is smooth. Generally speaking, if there is a unit root, the indicator is not smooth, and vice versa is smooth. As shown in Table 3, the Z value of each indicator is less than 0.05, indicating that the null hypothesis is not rejected, indicating that each indicator is smooth.

### 5.2. Benchmark Regression Results

Through the regression of the benchmark model, the impact of green technology transfer and environmental regulation on the regional green development chasm has been preliminarily verified. As shown in Table 4, model (1) is the benchmark model, model (2) and model (4) add control variables, respectively, model (3) and model (4) control the time fixed effect and the regional fixed effect. The regression results show that without considering the control variables, the impact of green technology transfer and environmental regulation on the regional green development chasm is significantly positive at the 1% level, and the significance remains unchanged after controlling for time fixed effects and regional fixed effects., the regression coefficients decrease after adding the control variable, but are still significant at the 1% level. Therefore, the results are somewhat robust, and the impact of green technology transfer and environmental regulation on the green development chasm has been preliminarily verified.

### 5.3. Spatial Panel Regression Results

By the calculation of the global Moran index for the green development gap index from 2005 to 2019 (Table 5), it was found that except 2005 and 2006, the P values of Moran’s I were significantly negative. The Moran’s I decrease over time, reflecting the increasing spatial negative correlation. The above result indicates that there is a significant negative spatial correlation in the Yangtze River Delta green development gap index, which needs to be measured using a spatial econometric model.

In order to improve the accuracy of the regression results, it is necessary to further use the SAR, SEM, SAC, and SDM spatial panel models that can take into account the spatial correlation of economic activities across regions. Through the Hausman test, fixed effects were chosen for the spatial panel econometric model and the regression results are shown in Table 6, models (1), (3), (5), and (7) are the basic regression estimation results, and models (2), (4), (6), and (8) are the estimated results after controlling other variables. Additionally, the above models all have considered time fixed effects and regional fixed effects.

From the estimation results in Table 6, the coefficients of spatial term of all four spatial econometric models are significantly positive, indicating that the local green development gap index is weighted by economic activities of neighboring regions. In terms of model fitting effect, compared with the SAR, SEM, and SAC models, the SDM model has the largest number of significant coefficients, indicating that the SDM model has the best fitting effect, the SDM model hence is chosen for analysis. It can be seen from the models (7) and (8) that the coefficients of green technology transfer (*GTT*) are all significantly negative, reflecting its positive effect on narrowing the green development chasm in the Yangtze River Delta. Differences in regional development are rooted in the divide of endowments, and it is difficult for less developed regions to narrow the development chasm without actor accumulation and technological progress. Studies show that the technological divide determines the gap in economic development levels between regions, which is also the key to achieving regional catch-up [84]. Technology transfer improves the efficiency of technology resource allocation in the Yangtze River Delta, which will help to promote the enhancement of resource endowments and technology levels in underdeveloped regions, enhance green development capabilities, and promote the convergence level in the Yangtze River Delta. The coefficient of environmental regulation (*ER*) is also significantly negative, reflecting that environmental regulation helps narrow the regional green development chasm. On the one hand, the implementation and severity of environmental regulation policies in the Yangtze River Delta are increasing, and the marginal cost of polluting enterprises is constantly approaching the marginal cost of society. Considering the long-term strong environmental regulation expectations and development interests, enterprises will increase their investment in green technology innovation, improve their green technology innovation capabilities, which in turn directly leads to the improvement of green development. On the other hand, with the deepening of regional integration in Yangtze River Delta, environmental policies and standards have continued to develop towards convergence, effectively reducing the occurrence of pollution transfer and narrowing the regional gaps of regional green technology innovation and green development. What needs to be emphasized is that in model (7) and model (8), the coefficient of environmental regulation is greater than that of green technology transfer, reflecting that the more positive role of environmental regulation on narrowing the green development chasm.

Model (8) adds the corresponding control variables, respectively. As a result, the technology absorptive capacity (*TAC*) is significantly negative at the 5% level, reflecting its negative impact on the green development chasm. This may be because the stock of green technology patents, which characterizes technology absorbing capacity, is highly concentrated in economically developed cities, while local green technology resources in less developed regions are weak and have a relatively limited role in supporting local green development. Government Science and Technology Investment (GSTI) is significantly negative at the 10% level, reflecting its positive effect on narrowing the green development chasm. As a supplier of public goods, the government science and technology investment level mirrors the government’s intervention in the market. Government spending on science and technology will, to some extent, compensate to the lack of market investment in green technology innovation. Energy consumption (*EC*) is significantly positive at the 5% level, echoing its negative effect on narrowing the green development chasm. Energy consumption is the main source of carbon emissions and a key factor in achieving green development. The energy consumption of production and living in the Yangtze River Delta is highly concentrated in economically developed cities, while energy efficiency and GDP per unit of energy consumption are significantly higher than those in underdeveloped cities, which are under greater pressure on energy conservation and emission reduction, resulting in an aggravating the gap of the inter-regional green development chasm. *GDP* per capita (*PGDP*) and industrial structure (*IND*) are not significant in the model.

From the models (9) and (10), the interaction between green technology transfer (*GTT*) and environmental regulation (*ER*) has a significant negative impact, which indicates that the combination of green technology transfer based on strong market and environmental regulation based on strong government has a positive effect on narrowing the regional green development chasm. For this reason, while promoting the transfer of green technology, it is necessary to strengthen the role of the government. Through the combination of environmental regulation and green technology transfer, the green technology innovation capacity of less developed regions can be enhanced, and the chasm with that of developed regions developed areas can be reduced.

### 5.4. Robustness Check

The results of the benchmark regression and the spatial panel regression are basically consistent, which shows the good robustness of the regression results. In order to further verify whether the effects of green technology transfer and environmental regulation on the green development chasm are robust, the explained variables are replaced by the industrial green development gap index (*IGDG*), which is calculated that the maximum value of the industrial added value of smoke and dust emissions per unit of each city in the same period is divided by the industrial added value of each city’s unit smoke and dust emission in the same period. At the same time, the total transfer of green technology (*GTT_total_*), which includes the amount of patent transfer in and out, is used to replace the explanatory variable, the amount of green technology transfer (*GTT*). Additionally, the models considered time fixed effects and regional fixed effects. The robustness test results are shown in the following Table 7. From the models (1) to (4), the significance and direction are basically unchanged, indicating that the results of the benchmark regression and spatial panel regression are robust.

### 5.5. Further Sub-Regional Inspection

The important purpose of green technology transfer across regions is to narrow the regional green development chasm. Therefore, it is necessary to analyze the heterogeneous influence of green technology transfer, environmental regulation, and green development chasm under different economic development levels. The outline of the integrated regional development of the Yangtze River Delta divides 41 cities into central and non-central areas, including 27 and 14 cities, respectively, with large gap in economic development situation. In 2019, the per capita GDP of the central group is about twice more than that of the non-central group. Accordingly, this paper divides the samples into two categories or regression analysis tests. Additionally, the models have considered time fixed effects and regional fixed effects. The Subregional spatial panel regression results are shown in the following Table 8. From model (1) and model (2), it is obvious that green technology transfer, environmental regulation, and the interaction have positive effects on narrowing the green development chasm in the central area, with a relatively frequently elements flow and close institutional policy among the 27 cities, the cross-regional green technology transfer and environmental regulation contribute effectively to the narrowing of the green development chasm in the central area. As seen from model (3) and model (4), except for the environmental regulation in model (3), the coefficients of green technology transfer and environmental regulation are generally insignificant, reflecting that green technology transfer and environmental regulation have failed to narrow the green development chasm of the non-central area in the Yangtze River Delta. The reason may be that due to the agglomeration of innovative elements and the market behavior characteristics of green technology transfer, the non-central area of the Yangtze River Delta is relatively marginalized in the green technology transfer network, and it is difficult to fully obtain green technology transfer, while the current environmental regulation cannot meet the needs of non-central areas to achieve high-quality development and narrow the gap with central areas.

## 6. Conclusions and Discussion

### 6.1. Conclusions

In response to global climate change, both the world and China have spared no efforts in promoting green technology transfer cooperation and strengthen environmental regulations in order to narrow the green development chasm and achieve the common goal of green development. However, both globally and in China, the green development chasm is always difficult to break through. The mechanism and effect of green technology transfer and environmental regulation on the green development chasm are worthy of in-depth analysis and examination. Based on neoclassical growth theory, network capital theory and environmental regulation theory, this paper incorporates green technology transfer, environmental regulation, and green development chasm into one identical framework, and then constructs a spatial econometric model of green technology transfer and environmental regulation affecting the green development chasm. By using green technology patent transfer data and panel data of 41 cities in the Yangtze River Delta from 2005 to 2019, this paper measured the change of the green development gap index, and then analyze the impact of green technology transfer and environmental regulation on the regional green development chasm.

The main conclusions are as follows: (1) Although the green development gap index in Yangtze River Delta has shown a W-shaped trend and the green development chasm is still obvious in recent years, during the empirical analysis in the benchmark model, spatial econometric model and robustness test, the coefficients of green technology transfer and environmental regulation are significantly negative, indicating that green technology transfer and environmental regulation play a positive role in narrowing the green development chasm. (2) The superposition of green technology transfer and environmental regulation significantly contributes to narrowing the green development chasm. Regional green development is highly dependent on the synergy of multiple factors and areas such as green technology, factor flow, and government regulation. The combination of two forces, green technology transfer and environmental regulation, can to a certain extent compensate for the deficiencies of government regulation and market regulation, forming a “joint force” to promote green development and narrows the green development chasm. (3) Regional heterogeneity plays a vital role on the process of regional green development and the reduction in chasm. In relatively developed regions, green technology transfer, environmental regulation and their interaction terms have a positive effect on narrowing the green development chasm. However, in economic relatively underdeveloped areas, only the interaction between green technology transfer and environmental regulation has a significant positive effect. Green technology transfer is highly agglomerated, and less developed regions are particularly marginalized in the green technology transfer network, which is difficult to narrow the green development chasm with developed regions by relying only on spontaneous technology transfer and government environmental regulation. It is necessary to strengthen the coordination between green technology transfer and environmental regulation in less developed regions.

### 6.2. Research Implications

In theory, different from the existing fragmentation research, this paper integrates green technology transfer, environmental regulation, and green development chasm, so that the research has a more holistic view. In fact, factor flow and government regulation are the areas of discussion in neoclassical economics and institutional economics, respectively. This paper highlights the need to discuss how to narrow the green development chasm under the goal of carbon neutrality based on multidisciplinary approach. On the basis of existing divergent views, this paper supports the view that green technology transfer and environmental regulation have positive effect on narrowing the green development chasm. At the same time, this paper emphasizes the need to dialectically examine the impact of green technology transfer and environmental regulation on the green development chasm. Neither market nor government, and factors nor institutions can effectively narrow the green development chasm. Additionally, green technology transfer must be overlaid with environmental regulation to effectively contribute to narrowing the green development chasm. Meanwhile, this paper also identifies the role of regional heterogeneity on green technology transfer and environmental regulation in narrowing the regional green development chasm.

In practice, achieving carbon neutrality goals and narrowing the green development chasm is a complex challenge and a systematic task. Both the world and China need to intensify efforts to transfer green technologies to underdeveloped regions, which is crucial to avoid development lock-in in underdeveloped regions. As an important provider of public goods, the government should play a more important role in the green technology transfer to avoid market failure in green technology transfer, meanwhile participating in market-incentive environmental regulation and command control environmental regulation such as environmental taxes and financial subsidies to promote green technology transfer and narrow the green development chasm. As China’s regional development varies greatly, narrowing the green development chasm through green technology transfer and environmental regulation cannot be “one size fits all”. Local environmental factors, including economic development level, technology absorption capacity, industrial structure, etc., need to be taken into account to make green technology transfer and environmental regulation more effective. Simultaneously, although China has issued national and local policies on promoting green technology innovation and technology transfer, most of them are scattered in various departmental documents, and a package of comprehensive support policies to green technology transfer, especially for underdeveloped areas, is still lacking. Hence, systematic support policies need to be formulated in China, to promote green technology transfer and narrow the regional green development chasm.

### 6.3. Policy Recommendations

Based on the research conclusions, this paper puts forward the following policy suggestions: Firstly, it is recommended to formulate support policy program for green technology transfer at the national or the Yangtze River Delta level to systematically promote green technology transfer. Additionally, Yangtze River Delta should accelerate the promotion and application of green technologies in cities with low technological level, laggard industrial structure, and high pressure on carbon emission reduction, so as to help them get rid of low-tech lock-in and high-carbon development lock-in. Secondly, focusing on green technology transfer needs and current restrictions, Yangtze River Delta should actively improve the green technology transfer cooperation ecosystem, including establishing cross-regional green technology transfer cooperation platforms or agencies, supporting existing platforms such as the Green Technology Bank of China to strengthen green technology transfer, encouraging the establishment of green technology innovation and guidance funds for the transformation of scientific and technological achievements, as well as exploring a benefit-sharing mechanism for the cross-regional transformation. Thirdly, the Yangtze River Delta should establish and implement strict environmental regulations, standards, systems, etc., and improve policies such as environmental taxes and pollution discharge fees, and rely on environmental regulations to promote the withdrawal of high-energy-consuming and high-polluting enterprises in underdeveloped areas. Moreover, considering the requirements of integrated regional development, it is necessary to accelerate the establishment of unified environmental regulation system to avoid the formation of “policy hollows” and pollution transfer, and to promote the overall realization of green development in Yangtze River Delta. Finally, a comprehensive system of policies and regulations, standards and norms, government planning, and markets should be established to promote multi-departmental linkage and multi-sectoral collaboration, so as to promote the realization of the overall goals of carbon peaking, carbon neutrality, and green development in the Yangtze River Delta.

### 6.4. Limitations and Future Lines of Research

This paper discusses the impact of green technology transfer and environmental regulation on the regional green development chasm, but there are still some limitations. Firstly, patent is an effective form to characterize green technology transfer, and it is helpful for quantitative analysis. However, technology transfer is a complex spatial process of technological flow, including talent flow, technology alliance, commodity trade, direct investment, technical assistance, industry-university-research cooperation, etc. These data are not used in this study due to the obtaining difficulty of completeness. Secondly, the measurement of environmental regulation and green development chasm is also a complex system, and it is difficult to have a perfect indicator to describe it well, which inevitably makes the conclusions of this paper inaccurate to a certain extent. Thirdly, enterprises are important main bodies of green technology transfer and important implementation objects of environmental regulation. This paper mainly analyzes from a regional perspective, and does not consider enough the role of green management and lean green in the green technology transfer, environmental regulation, and green development chasm. Finally, The Yangtze River Delta is a region with relatively high level of economic development, active technology transfer and strong environmental regulation in China. Empirical tests based on the Yangtze River Delta may not reflect general patterns in China or globally.

The following issues are worthy of further research: on one hand, future research can obtain diversified green technology transfer data such as talent flow and industry–university–research cooperation through enterprise surveys, and then analyze the role of enterprise green management in green technology transfer, environmental regulation, and green development. On the other hand, in terms of the research subjects, future research can expand the research sample to the whole country or even conduct a global analysis, and then, compare and analyze the characteristics of the impact of technology transfer and environmental regulation on the regional green development chasm in different types and development levels.

## Figures and Tables

**Figure 1 ijerph-19-13925-f001:**
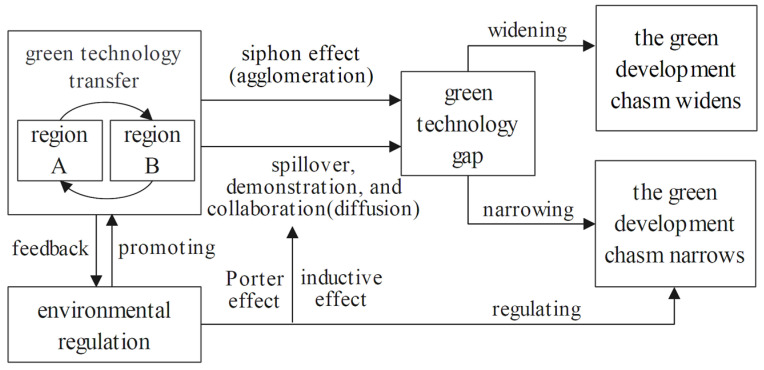
The mechanism between green technology transfer, environmental regulation, and green development chasm.

**Figure 3 ijerph-19-13925-f003:**
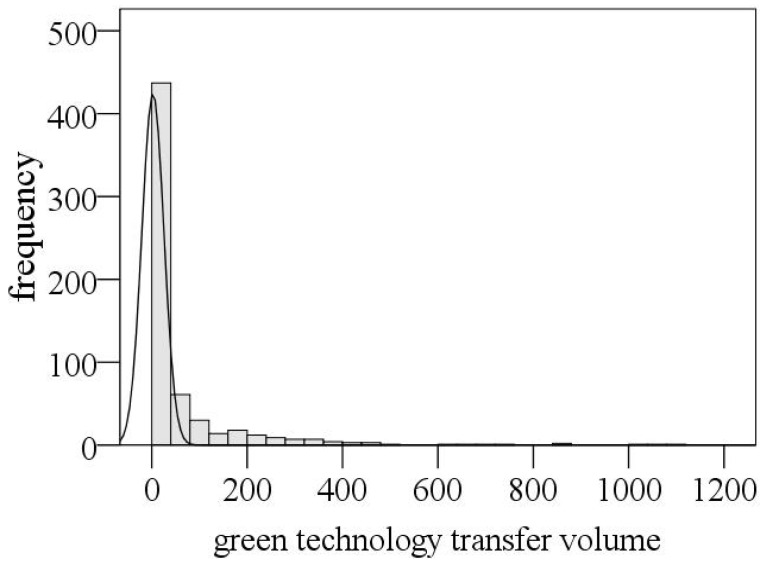
Frequency distribution of green technology transfer in the Yangtze River Delta from 2005 to 2019.

**Table 1 ijerph-19-13925-t001:** Variable selection and corresponding explanation.

Variable	Indicator Explanation	Indicator Nature	Variable Name
Green Development Gap Index	The gap between the total GDP per unit carbon emission in each region and the highest level in the same period	explained variable	GDG
Green Technology Transfer	Number of green technology patents transferred in	explanatory variables	GTT
Environmental Regulation	Environmental regulation index	explanatory variables	ER
GDP per capita	GDP per capita	control variable	PGDP
technology Absorptive capacity	Green technology patent stock	control variable	TAC
Industrial structure	The proportion of secondary industry	control variable	IS
Government Science and Technology Investment	Government finance investment on science and technology	control variable	GSTI
Energy Consumption	Total electricity consumption of the society	control variable	EC

**Table 2 ijerph-19-13925-t002:** Descriptive statistics for variables.

Variable	Variable Name	Mean	SD	Min	Max	N
Green Development Gap Index	GDG	5.14	5.04	1	33.58	615
Green Technology Transfer	GTT	120.20	260.96	0	2220	615
environmental regulation	ER	0.82	0.08	0.52	0.95	615
GDP per capita	PGDP	54,520.12	38,054.23	3761	199,017	615
technology absorptive capacity	TAC	79.26	161.26	0	1213	615
Industrial structure	IS	48.74	8.57	23.97	74.73	615
Government technology investment	GSTI	3.91	4.64	0.04	28.74	615
Energy consumption	EC	162.61	255.91	3.77	1568.6	615

**Table 3 ijerph-19-13925-t003:** LM unit root test results.

Variable	GDG	GTT	ER	PGDP	TAC	IS	GSTI	EC
Statistic	7.828	8.698	14.281	7.962	8.069	9.556	11.540	3.218
Z	0.000	0.000	0.000	0.000	0.000	0.000	0.000	0.000

**Table 4 ijerph-19-13925-t004:** Benchmark regression results.

Variable	Model (1)	Model (2)	Models (3)	Models (4)
lnGTT	−0.042 ***	−0.110 ***	−0.021 ***	−0.019 ***
	(0.013)	(0.023)	(0.009)	(0.009)
lnER	−0.248 ***	0.176	−0.674 ***	−0.618 ***
	(0.328)	(0.139)	(0.182)	(0.185)
control variable		control		control
time fixed effects			control	control
city fixed effect			control	control
N	615	615	615	615

Note: ***, **, * indicate significance at the 1%, 5% and 10% levels, respectively, and the standard errors are in brackets.

**Table 5 ijerph-19-13925-t005:** Moran‘s I index of green economy growth performance over the years.

Years	Moran’s I	Z-Value	*p*-Value
2005	−0.032	−0.499	0.309
2006	−0.032	−0.529	0.298
2007	−0.042 *	−1.088	0.099
2008	−0.041 *	−1.214	0.083
2009	−0.042 *	−1.131	0.089
2010	0.038 *	−1.357	0.069
2011	0.033 *	−1.177	0.082
2012	−0.038 *	−1.294	0.060
2013	−0.045 **	−1.721	0.044
2014	−0.047 **	−1.807	0.034
2015	−0.054 **	−2.158	0.015
2016	−0.049 **	−1.765	0.039
2017	−0.053 **	−2.041	0.021
2018	−0.051 ***	−2.917	0.008
2019	−0.046 ***	−3.559	0.005

Note: ***, **, * indicate significance at the 1%, 5% and 10% levels, respectively, and the standard errors are in brackets.

**Table 6 ijerph-19-13925-t006:** Spatial panel regression results.

Variable	SAR		SEM		SAC		SDM			
	Model (1)	Model (2)	Model (3)	Model (4)	Model (5)	Model (6)	Model (7)	Model (8)	Model (9)	Model (10)
lnGTT	−0.025 ***	−0.037 ***	−0.024 ***	−0.028 ***	−0.019 **	−0.019 **	−0.020 **	−0.016 *	−0.181 **	−0.203 *
	(0.006)	(0.008)	(0.008)	(0.009)	(0.008)	(0.009)	(0.009)	(0.009)	(0.063)	(0.065)
lnER	−0.662 ***	−0.679 ***	−0.674 ***	−0.552 ***	−0.632 ***	−0.558 ***	−0.679 ***	−0.579 ***	−0.836 ***	−0.790 ***
	(0.125)	(0.162)	(0.155)	(0.160)	(0.153)	(0.154)	(0.162)	(0.168)	(0.170)	(0.179)
lnGTT × lnE*R*									−0.238 ***	−0.258 ***
									(0.075)	(0.076)
lnPGDP		0.001		0.007		0.013		0.013		0.015
		(0.012)		(0.009)		(0.011)		(0.012)		(0.011)
lnTAC		0.012		0.024 **		0.030 ***		0.031 ***		0.028 **
		(0.010)		(0.011)		(0.011)		(0.011)		(0.011)
lnIS		0.029		0.031		0.017		0.024		0.012
		(0.031)		(0.032)		(0.030)		(0.032)		(0.032)
lnGSTI		0.008		−0.034		−0.034		−0.048 *		−0.057 *
		(0.012)		(0.026)		(0.025)		(0.026)		(0.026)
lnEC		0.024 **		0.024 *		0.024 **		0.029 **		0.031 **
		(0.012)		(0.013)		(0.012)		(0.017)		(0.013)
rho	0.765 ***	0.748 ***	0.770 ***	0.780 ***	−1.081 ***	−1.082 ***	0.771 ***	0.694 ***	0.760 ***	0.694 ***
	(0.039)	(0.042)	(0.040)	(0.040)	(0.327)	(0.312)	(0.039)	(0.056)	(0.041)	(0.056)
sigma2_e	0.027 ***	0.027 ***	0.029 ***	0.029 ***	0.027 ***	0.026 ***	0.027 ***	0.026 ***	0.026 ***	0.026 ***
	(0.002)	(0.002)	(0.001)	(0.002)	(0.002)	(0.001)	(0.002)	(0.001)	(0.001)	(0.001)
time fixed effects	control	control	control	control	control	control	control	control	control	control
city fixed effect	control	control	control	control	control	control	control	control	control	control
N	615	615	615	615	615	615	615	615	615	615

Note: ***, **, * indicate significance at the 1%, 5% and 10% levels, respectively, and the standard errors are in brackets.

**Table 7 ijerph-19-13925-t007:** Robustness test results.

Variable	The Impact of Green Technology Transfer and Environmental Regulation on the Gap of Industrial Green Development		Green Technology Transfer and Environmental Regulation on the Green Development Gap Index	
	Model (1)	Model (2)	Models (3)	Models (4)
lnGTT	−0.551 ***	−0.561 ***		
	(0.140)	(0.144)		
lnGTT_total_			−0.424 ***	−0.423 ***
			(0.125)	(0.129)
lnER	−0.757 **	−0.633 *	−0.635 *	−0.510 *
	(0.378)	(0.358)	(0.388)	(0.313)
lnGTT × lnER	−0.660 **	−0.668 ***	−0.486 **	−0.484 **
	(0.164)	(0.169)	(0.151)	(0.156)
control variable		control		control
rho	0.540 ***	0.370 **	0.563 ***	0.334 ***
	(0.074)	(0.120)	(0.070)	(0.124)
sigma2_e	0.128 ***	0.127 ***	0.130 ***	0.129 ***
	(0.007)	(0.007)	(0.007)	(0.007)
time fixed effects	control	control	control	control
city fixed effect	control	control	control	control
N	615	615	615	615

Note: ***, **, * indicate significance at the 1%, 5% and 10% levels, respectively, and the standard errors are in brackets.

**Table 8 ijerph-19-13925-t008:** Subregional spatial panel regression results.

Variable	Central District		Non-Central Areas	
	Model (1)	Model (2)	Models (3)	Models (4)
lnGTT	−0.147 **	−0.248 ***	−0.040	−0.127
	(0.071)	(0.075)	(0.158)	(0.159)
lnER	−0.745 ***	−0.828 ***	−0.891 ***	−0.166
	(0.236)	(0.244)	(0.272)	(0.320)
lnGTT × ln ER	−0.205 **	−0.326 ***	−0.241 *	−0.151 *
	(0.108)	(0.088)	(0.136)	(0.086)
control variable		control		control
rho	0.706 ***	0.539 ***	0.603 ***	0.558 ***
	(0.051)	(0.085)	(0.066)	(0.073)
sigma2_e	0.028 ***	0.025 ***	0.028 ***	0.024 ***
	(0.002)	(0.002)	(0.003)	(0.002)
time fixed effects	control	control	control	control
city fixed effect	control	control	control	control
N	405	210	405	210

Note: ***, **, * indicate significance at the 1%, 5% and 10% levels, respectively, and the standard errors are in brackets.

## Data Availability

Not applicable.

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
