# Peer review of "Green Technology Transfer, Environmental Regulation, and Regional Green Development Chasm: Based on the Empirical Evidence from Yangtze River Delta"

_ijerph, 2022, doi:10.3390/ijerph192113925_

Round 1

Reviewer 1 Report

In the introduction, there is a need to support with the appropriate references ( institutions or organizations) the initial statements. 

When mentioning "a large number of scholars" more than 2 references are required to illustrate the broad diffusion of the argument. 

Between lines 67 and 88 there is complete absence of referencing - needs to be corrected.

The literature review must not concentrate only on the Chinese case. It is important to grasp this evidence, however in comparative terms not in descriptive terms (perhaps this part should be 2.2 - the second half of the literature review. In this vein, the conceptual model presented in section 3 should be the first part of the literature review, as the general arguments are provided. Also, in this part references are missing.

When the authors mention "some scholars" additional references need to be included in the document. 

In what relates to the conceptual framework scheme, the authors mention regulation only, however, the can be an incentive through fiscal benefits such as grants, subventions or lowering tax rates...It is important to balance policy instruments - the comparison between both sides should be taken into consideration in the paper. I suggest taking into consideration the evidence from Carrots or Sticks: Which Policies Matter the Most in Sustainable Resource Management? Resources 2021, 10, 12. https://doi.org/10.3390/resources10020012.

In the spatial model there is need to enlighten the reader about how time is taken into consideration. 

Endogenous and exogenous variables are more used terminology than the presented ones - I suggest changing. 

Average value should be mean instead. 

Please confirm the contents of table 5 as some years appear repeatedly and no distinction is made between the first and the second. 

It is mandatory to contextualize the present findings with the previous to discuss if the findings go in line with previous research or not. Also, what are the "surprises" brought to light by the research? How far does the study contribute to theory or to practice? 

In sum, is the current policy package accurate, or do the new findings point elsewhere?

Best of luck with your research!

Author Response

Dear Reviewer:

Thank you for your comments concerning our manuscript. Those constructive comments are all valuable and very helpful for revising and improving our paper, as well as the important guiding significance to our research. We have studied comments carefully and have made revises which we hope meet with approval. And “Track Changes” function is used for all revises in the manuscript. For accurate answer, we replied each comment respectively.

Comment: In the introduction, there is a need to support with the appropriate references ( institutions or organizations) the initial statements.

When mentioning "a large number of scholars" more than 2 references are required to illustrate the broad diffusion of the argument.

Between lines 67 and 88 there is complete absence of referencing - needs to be corrected.

Authors’ Response: Thanks a lot for reminding us of this important point. We have supplemented some more references when mentioning “a large number of scholars”; meanwhile we have added the appropriate reference between lines 67 and 88, you can see the more details in the introduction and reference part of revised manuscript.

Comment: The literature review must not concentrate only on the Chinese case. It is important to grasp this evidence, however in comparative terms not in descriptive terms (perhaps this part should be 2.2 - the second half of the literature review. In this vein, the conceptual model presented in section 3 should be the first part of the literature review, as the general arguments are provided. Also, in this part references are missing.

Authors’ Response: Thanks a lot for reminding us of these important points. Firstly, we supplemented more foreign cases and studies to the literature review. Secondly, we try our best to use more comparative and inductive terms in the literature review. Thirdly, the writing logic of this paper is to critically review the existing relevant research and summarize the theoretical research framework, which is focus on mechanism of green technology transfer and environmental regulation on regional green development chasm (that is, how and what effect it has), hence we think it may be better if we put the theoretical mechanism after the literature review. Meanwhile we have further revised and deepened the original theoretical framework and its related statements, as shown in the section 3 of the revised manuscript. Finally, we have added a large number of references, taking into account the previous comment.

Comment: When the authors mention “some scholars” additional references need to be included in the document.

Authors’ Response: Thanks a lot for reminding us of this important point. We have modified this in the relevant section, you can find more details in the revised manuscript.

Comment: In what relates to the conceptual framework scheme, the authors mention regulation only, however, they can be an incentive through fiscal benefits such as grants, subventions or lowering tax rates...It is important to balance policy instruments - the comparison between both sides should be taken into consideration in the paper. I suggest taking into consideration the evidence from Carrots or Sticks: Which Policies Matter the Most in Sustainable Resource Management? Resources 2021, 10, 12. https://doi.org/10.3390/resources10020012.

Authors’ Response: Thanks a lot for reminding us of these important points. In the revised manuscript, we add some related argument between line 259 and 268. It is important to note, however, that we agree that there are significant differences and pathways between different environmental regulatory policies, such as command-and-control type and market-incentive type, but both are normally interactive and irreplaceable, and the series of criteria and standards for cohesion is commonly employed as the architecture of environmental regulation. We hope that future studies will be able to compare these two in more detail and thank you very much for providing us with new research ideas.

Comment: In the spatial model there is need to enlighten the reader about how time is taken into consideration.

Authors’ Response: Thanks a lot for reminding us of these important points. Both time fixed effects and spatial fixed effects are taken into account when we construct OLS models, spatial econometric models, and stata15.0 operations. We have taken reference from some prior papers and have not displayed the fixed effects in the paper for reasons of page limitation. I am sorry about the omission. In the revised version, we have put fixed effects in Tables 6, 7 and 8 with explanations for the sake of higher completeness and standardization.

Comment: Endogenous and exogenous variables are more used terminology than the presented ones

Authors’ Response: Thanks a lot for reminding us of these important points. Referred to a large number of papers (e.g. [13], [27], [30], [33]), we found that the SDM model and other spatial econometric models usually do not consider endogenous variables and exogenous variables. At the same time, the main purpose of this paper is to explain the impact of two explanatory variables, green technology transfer and environmental regulation, on the explained variable regional green development chasm. So, we use the terms explained variable, explanatory variable, and control variable. Please correct us if our understanding is wrong.

Comment: Average value should be mean instead.

Authors’ Response: Thanks a lot for reminding us of this important point, we have corrected it in the revised manuscript.

Comment: Please confirm the contents of table 5 as some years appear repeatedly and no distinction is made between the first and the second.

Authors’ Response: Thanks a lot for reminding us of this important point. This was our mistakes and we have corrected it.

Comment: It is mandatory to contextualize the present findings with the previous to discuss if the findings go in line with previous research or not. Also, what are the "surprises" brought to light by the research? How far does the study contribute to theory or to practice?

Authors’ Response: Thanks a lot for reminding us of this important point. We compared our results with previous studies, and then proposed our theoretical and practical contributions, as shown in “6.2. Research Implications”.

In theory, different from the existing fragmentation research, this paper integrates green technology transfer, environmental regulation and green development chasm, so that the research has a more integrated view. And this paper supports that green technology transfer and environmental regulation have positive effect on narrowing the green development chasm, and green technology transfer must be interactive with environmental regulation in order to effectively contribute to narrowing the green development chasm. This paper also notices that there is regional heterogeneity in the role of green technology transfer and environmental regulation in narrowing the regional green development chasm.

In practice, both the world and China need intensify efforts to transfer green technologies to underdeveloped regions, which is the key measures to avoiding lock-in in underdeveloped regions. Narrowing the green development chasm through green technology transfer and environmental regulation cannot be “one size fits all”. China needs to formulate systematic support policies, promote green technology transfer in an organized manner to narrow the regional green development chasm.

Comment: In sum, is the current policy package accurate, or do the new findings point elsewhere?

Authors’ Response: Thanks a lot for reminding us of this important point. We have noticed that there is still a lack of policies package for green technology transfer in the Yangtze River Delta, a lack of incentives for green technology transfer, and fragmented technology transfer and environmental regulations are still relatively. To this end, we put forward some corresponding suggestions. E.g., it is recommended to formulate support policies for green technology transfer at the national or the Yangtze River Delta to systematically promote green technology transfer and make green technology transfer more scientific and efficient. Focusing on green technology transfer needs and current bottlenecks, Yangtze River Delta should actively improve the green technology transfer cooperation ecosystem.

We will be happy to revise the manuscript further based on helpful comments from the reviewer.

Best regards!

Reviewer 2 Report

The Abstract of the article is interesting, however it should be improved by the authors. It is recommended that the authors, in the final part of the Abstract, include some of the results obtained with this research.

[line 44-45] The authors state "However, green technology innovation is characterised by multiple investments, high risks, long cycle, as well as double externalities...", yet why do the authors, in addition to "green technologies", not address green management (lean green) which is being adopted by several companies? The inclusion of this topic in the paper, will strongly enhance the quality and framework of the research presented in this paper by the authors. I recommend that the authors consult the article "Lean Green—The Importance of Integrating Environment into Lean Philosophy—A Case Study" (https://doi.org/10.1007/978-3-030-41429-0_21).

[line 145-147] The authors in the article state the following: “Economically developed regions can be more likely to avoid the negative impact of environmental regulation, and environmental regulation will even promote the agglomeration of innovation factors towards to developed regions”. The authors quite rightly introduce the economic side of the article, along with "green technologies". However, why don't the authors include "sustainability" in their Literature Review? As the three pillars of sustainability are: economic, environmental, and social, it would be very good to include this topic. To help the authors I can recommend the articles: " Combining lean and green practices to achieve a superior performance: The contribution for a sustainable development and competitiveness—An empirical study on the Portuguese context " (https://doi.org/10.1002/csr.2242) and " Barriers to the adoption of green operational practices at Brazilian companies: effects on green and operational performance " (https://doi.org/10.1080/00207543.2016.1154997).

It would be very good if the authors would identify what the Research Question, of this article is. What do the authors want to demonstrate by publishing this research? What was the gap that the authors identified in their literature review? This information introduced in the article would allow the authors to clearly justify the need and importance of publishing this article.

In lines 426-428, you mention which methodology was used by the authors in this research, correct? This information should appear in the explanation of the methodology and be further explained to facilitate the understanding of this research by the scientific community.

It would be very important that the authors at the end of the article indicate what future work can be developed from this research. It would be very good if this research did not stop here, given its potential, and that the scientific community could give it continuity, and give it the importance and relevance that this research may have in the future.

The authors should add the limitations that this research had. What did the authors want to have done during this research, and for whatever reason, it was not possible? What was or what were the difficulty/limitations that the authors felt and experienced? This information would be very useful to share, to help future research, and help interpret some of the results obtained.

Author Response

Dear Reviewer:

Thank you for your comments concerning our manuscript. Those constructive comments are all valuable and very helpful for revising and improving our paper, as well as the important guiding significance to our research. We have studied comments carefully and have made revises which we hope meet with approval. And “Track Changes” function is used for all revises in the manuscript. For accurate answer, we replied each comment respectively.

Comment: The Abstract of the article is interesting, however it should be improved by the authors. It is recommended that the authors, in the final part of the Abstract, include some of the results obtained with this research.

Authors’ Response: Thanks a lot for reminding us of this important point. We have revised the abstract and highlighted the results of this research.

Comment: [line 44-45] The authors state “However, green technology innovation is characterised by multiple investments, high risks, long cycle, as well as double externalities...”, yet why do the authors, in addition to "green technologies", not address green management (lean green) which is being adopted by several companies? The inclusion of this topic in the paper, will strongly enhance the quality and framework of the research presented in this paper by the authors. I recommend that the authors consult the article "Lean Green—The Importance of Integrating Environment into Lean Philosophy—A Case Study" (https://doi.org/10.1007/978-3-030-41429-0_21).

Authors’ Response: Thanks a lot for reminding us of this important point and recommended paper. Green management (lean green) is indeed important for sustainable development, and it is also a very important supplement to our research. We have added green management to the literature review and theoretical mechanisms. Since this paper mainly studies from a regional perspective and conducts empirical analysis based on urban panel data, it does not consider much from the micro perspective of enterprises. Of course, this suggestion is of great importance to us, we also put green management into the limitations and may be the focus of our next study.

Comment: [line 145-147] The authors in the article state the following: “Economically developed regions can be more likely to avoid the negative impact of environmental regulation, and environmental regulation will even promote the agglomeration of innovation factors towards to developed regions”. The authors quite rightly introduce the economic side of the article, along with "green technologies". However, why don't the authors include “sustainability” in their Literature Review? As the three pillars of sustainability are: economic, environmental, and social, it would be very good to include this topic. To help the authors I can recommend the articles: “Combining lean and green practices to achieve a superior performance: The contribution for a sustainable development and competitiveness—An empirical study on the Portuguese context” (https://doi.org/10.1002/csr.2242) and “Barriers to the adoption of green operational practices at Brazilian companies: effects on green and operational performance” (https://doi.org/10.1080/00207543.2016.1154997).

Authors’ Response: Thanks a lot for reminding us of this important point. The papers you have recommended are also valuable for us, we added some green practices and sustainability into the revised manuscript, and you can find the details between lines 170 and 179. We proposed that the impact of environmental regulation on the green development chasm is also reflected in the environmental and social dimensions. Related studies have found that environmental regulation reduces the transfer of pollution between regions and promotes the convergence of environmental efficiency between regions. Mean-while, some scholars have also discussed from the micro-level of enterprises Lean green and green management will motivate enterprises to adopt more environmentally friendly practices, adopt clean technologies, reduce resource consumption, and waste discharge, as well as waste in different links such as production, transportation, inventory, etc. In turn, they improve the sustainable performance of enterprises and enhance green competitive advantage, thereby breaking the green barrier. At the same time, environmental regulation also promotes the spatial spillover of ecological welfare, thereby narrowing the gap in ecological welfare between regions.

Comment: It would be very good if the authors would identify what the Research Question, of this article is. What do the authors want to demonstrate by publishing this research? What was the gap that the authors identified in their literature review? This information introduced in the article would allow the authors to clearly justify the need and importance of publishing this article.

Authors’ Response: Thanks a lot for reminding us of this important point. The following changes have been made to highlight the necessity and importance: First, in the introduction (Line 86-89), we propose the research question and the possible contributions. Second, we further identify the shortcomings of existing research in the section of literature review. Again, we supplement the theoretical and practical contributions of the paper in 6.2. E.g., most existing studies focus only on one single dimension, such as technology transfer and/or environmental regulation, on the green development gap, which may lead to fragmentation between theory and practice; the characteristics of the regional green development chasm and the role of green technology transfer and environmental regulation are still somewhat controversial, and the analysis of the mechanism o is still limited

Comment: In lines 426-428, you mention which methodology was used by the authors in this research, correct? This information should appear in the explanation of the methodology and be further explained to facilitate the understanding of this research by the scientific community.

Authors’ Response: Thanks a lot for reminding us of this important point. We supplemented the correlation between the data on electricity consumption and energy consumption, and the fitted result. In the paper, we compared the logarithm of the data of electricity consumption and energy consumption in various provinces and cities in China from 2005 to 2019, and the correlation coefficient reaches 0.892. At the same time, the scatter plot is shown below. To save space, we did not include scatterplots in the manuscript. We also add an explanation of the advantages of using electricity consumption data, which is approved by the prior studies ([75], [76], [77]).

Figure 1. Scatter plot of electricity consumption and energy consumption in various provinces and cities in China from 2005 to 2019.

Comment: It would be very important that the authors at the end of the article indicate what future work can be developed from this research. It would be very good if this research did not stop here, given its potential, and that the scientific community could give it continuity, and give it the importance and relevance that this research may have in the future.

Authors’ Response: Thanks a lot for reminding us of this important point. We proposed the potential directions of research in this field in 6.4. E.g., first, future research can obtain diversified green technology transfer data such as talent flow and industry-university-research cooperation through enterprise surveys, and then analyze the role of enterprise green management in green technology transfer, environmental regulation, and green development. Second, in terms of the research subjects, future research can expand the research sample to the whole country or even carry out global analysis, and then, compare and analyze the characteristics of technology transfer and environmental regulation on the regional green development chasm between regions in different types and development levels.

Comment: The authors should add the limitations that this research had. What did the authors want to have done during this research, and for whatever reason, it was not possible? What was or what were the difficulty/limitations that the authors felt and experienced? This information would be very useful to share, to help future research, and help interpret some of the results obtained.

Authors’ Response: Thanks a lot for reminding us of this important point. We supplement the deficiencies from research data, microscopic perspectives, and area limitation. E.g., green technology patent transfer is an effective measure to characterize green technology transfer, and it is helpful for quantitative analysis. However, technology transfer is a complex process of technological flow, including talent flow, technology alliance, commodity trade, direct investment, technical assistance, industry-university-research cooperation, etc. These data are not used for this study due to the difficulty of access. Meanwhile, the measurement of environmental regulation and green development chasm is also complicated, and it is difficult to have a perfect indicator to describe, which inevitably makes the conclusions of this paper inaccurate to some extent. Moreover, enterprises are the main bodies of green technology transfer and implementation objects of environmental regulation. This paper mainly analyzes from a regional perspective, and does not consider green management, lean green, and the role of enterprises in the green technology transfer, environmental regulation, and green development chasm. Additionally, the Yangtze River Delta is a region with relatively high economic development, proactive technology transfer and effective environmental regulation in China. Empirical analysis based on the Yangtze River Delta may not reflect general situation in China or worldwide.

We will be happy to revise the manuscript further based on helpful comments from the reviewer.

Best regards!

Round 2

Reviewer 1 Report

I would like to thank the authors for bringing a revised version of the article. The present version benefited from the accommodation of the comments provided. 

After reading, once more the article I believe that it is important to provide the reader with a careful definition/conceptualization of "green development chasm" in theoretical terms as well as its implications. Why does it worth considering this conceptual definition and not others? What is the purpose of fighting "regional green development chasm"?

As I previously mentioned, the document needs deep proofreading. some sentences are misleading. The scientific soundness also needs improvement. 

O would recommend further consolidating the policy recommendations section. Bring specific policy actions which will reduce the difference between the strong ecological innovators and the weak.

Best of luck!

Author Response

Dear Reviewer:

Thank you for your comments concerning our manuscript. Those constructive comments are all valuable and very helpful for revising and improving our paper, as well as the important guiding significance to our research. We have studied comments carefully and have made revises which we hope meet with approval. And “Track Changes” function is used for all revises in the manuscript. For accurate answer, we replied each comment respectively.

Comment: It is important to provide the reader with a careful definition/conceptualization of “green development chasm” in theoretical terms as well as its implications. Why does it worth considering this conceptual definition and not others? What is the purpose of fighting “regional green development chasm”?

Authors’ Response: Thanks a lot for reminding us of this important point. We have provided a careful definition of “green development chasm” between lines 114 and 118, where you can see the more details in the revised manuscript. The analysis is as follows:

Generally, chasm is similar as the concepts such as imbalance and disparity, which is widely used in discussions on digital chasm [18], innovation chasm [19], governance chasm [20], and ideological chasm [21]. In the context of global climate crisis, the green development chasm has become a hot topic in academia to discuss the inequalities in the global green economy level, green technology, and green system [22,23].

As for the reason for considering this conceptual definition and the purpose of fighting “regional green development chasm”, we have made the following analysis:

Despite the active emphasis on “inclusive green growth” by governments and international institutions, global green development inequalities are still rising [24]. A large number of scholars have called for bridging the regional green development chasm, which is also related to the smooth realization of the common goals of addressing global warming and green development[25,26].

Comment: The document needs deep proofreading. some sentences are misleading. The scientific soundness also needs improvement.

Authors’ Response: Thanks a lot for reminding us of these important points. We have tried to proofread the full text carefully to make the expression more standardized, especially the concept in Chinese context. Moreover, we are communicating with editors to provide language editing services offered by the journal, which is more effective to improve the language quality and readability.

Comment: I would recommend further consolidating the policy recommendations section. Bring specific policy actions which will reduce the difference between the strong ecological innovators and the weak.

Authors’ Response: Thanks a lot for reminding us of this important point. We have reorganized the part and proposed several more specific policy suggestions. Meanwhile, we have reduced the policy recommendations that are not directly relevant such as the difference between the strong and weak ecological innovators.

We will be happy to revise the manuscript further based on helpful comments from the reviewer.

Best regards!

Reviewer 2 Report

The authors have strongly improved the quality and scientific level of the article! Excellent work!

Author Response

Dear Reviewer:

Thank you so much for your comments concerning our manuscript.

Best regards!